# Comparative Metagenomic Analysis Reveals Rhizosphere Microbiome Assembly and Functional Adaptation Changes Caused by Clubroot Disease in Chinese Cabbage

**DOI:** 10.3390/microorganisms12071370

**Published:** 2024-07-04

**Authors:** Yong Liu, Jia Lai, Xiaofang Sun, Ling Huang, Yuzhen Sheng, Qianfang Zhang, Hualan Zeng, Yinchao Zhang, Pengsheng Ye, Shugu Wei

**Affiliations:** Industrial Crops Research Institute, Sichuan Academy of Agricultural Sciences/The Key Laboratory of Vegetable Germplasm and Variety Innovation in Sichuan Province, Chengdu 610300, China; liuyong918@126.com (Y.L.); laijia028@163.com (J.L.); sunxiaofang207@163.com (X.S.); b1983429@126.com (L.H.); shengyuzhen0516@126.com (Y.S.); zqfzzf@126.com (Q.Z.); zhl0529@126.com (H.Z.); yinchao1@126.com (Y.Z.)

**Keywords:** Chinese cabbage, clubroot disease, rhizosphere microbiome, metagenomic sequencing

## Abstract

Clubroot is a major disease and severe threat to Chinese cabbage, and it is caused by the pathogen *Plasmodiophora brassicae* Woron. This pathogen is an obligate biotrophic protist and can persist in soil in the form of resting spores for more than 18 years, which can easily be transmitted through a number of agents, resulting in significant economic losses to global Chinese cabbage production. Rhizosphere microbiomes play fundamental roles in the occurrence and development of plant diseases. The changes in the rhizosphere microorganisms could reveal the severity of plant diseases and provide the basis for their control. Here, we studied the rhizosphere microbiota after clubroot disease infections with different severities by employing metagenomic sequencing, with the aim of exploring the relationships between plant health, rhizosphere microbial communities, and soil environments; then, we identified potential biomarker microbes of clubroot disease. The results showed that clubroot disease severity significantly affected the microbial community composition and structure of the rhizosphere soil, and microbial functions were also dramatically influenced by it. Four different microbes that had great potential in the biocontrol of clubroot disease were identified from the obtained results; they were the genera *Pseudomonas*, *Gemmatimonas*, *Sphingomonas*, and *Nocardioides*. Soil pH, organic matter contents, total nitrogen, and cation exchange capacity were the major environmental factors modulating plant microbiome assembly. In addition, microbial environmental information processing was extremely strengthened when the plant was subjected to pathogen invasion, but weakened when the disease became serious. In particular, oxidative phosphorylation and glycerol-1-phosphatase might have critical functions in enhancing Chinese cabbage’s resistance to clubroot disease. This work revealed the interactions and potential mechanisms among Chinese cabbage, soil environmental factors, clubroot disease, and microbial community structure and functions, which may provide a novel foundation for further studies using microbiological or metabolic methods to develop disease-resistant cultivation technologies.

## 1. Introduction

Chinese cabbage is one of the indispensable vegetables in the life and daily diet of Chinese people, which has been in cultivation since the fifth century with its consumption ranking first among all vegetables [1]. Clubroot disease is a common soil-borne disease induced by *Plasmodiophora brassicas* Wor., which seriously damages cruciferous vegetables [2]. This disease was first reported 280 years ago [3], and it greatly influences the production of Chinese cabbage. According to previous investigation data, the clubroot disease incidence in Chinese cabbage has increased to over 60% in the last three years [4]. Chinese cabbage infected by clubroot disease has swollen roots, wilted leaves, and stationary growth, which seriously limits its yield and quality [5]. The geographical distribution of clubroot disease is wide, the physiological differences between strains are complex, and the dormant sporangium can live in the soil for a long time [6], which brings great challenges to the prevention and control of clubroot disease. Currently, there are many measures from chemical, biological, and agricultural aspects for the prevention and control of *P. brassicas* infection, which mainly include spraying chemical agents [7], rational crop rotation and fertilization [8], soil amendments [9], the application of biological agents and bio-organic fertilizers [10,11], and cultivating varieties with disease resistance [12]; all of these methods have achieved certain effective results but are insufficient for controlling the disease. Crop rotation can maintain the diversity of plant species in the ecosystem, improve the diversity and structure of rhizosphere microorganisms, and protect plants from biological stress [8]. Soil amendments can obviously promote the growth of Chinese cabbage and the formation of soil aggregates, which is essential for coordinating the soil environment, maintaining the stability of the soil structure, and maintaining soil moisture and nutrients [9]. Although chemical fungicides are effective in controlling soil-borne diseases and have a low cost, their long-term and extensive use will increase the agent resistance of pathogens and create problems such as pesticide residues and environmental pollution [13]. Therefore, there is an urgent need to mitigate this devastating disease by developing durable, efficient, and environmentally friendly control measures. Biological control is a specific, safe, effective, sustainable, and environmentally friendly method to control soil-borne diseases by inducing host resistance, inducing niche and nutrition competition, and regulating rhizosphere microecology through the use of the mutually beneficial relationship between biocontrol microbes and plants [14,15].

In order to develop stable and efficient biological control products or methods, we first need to understand the mechanism that rhizosphere microecology uses to control pathogens. The rhizosphere is a narrow region of soil or substrate known as the root microbiome, which is directly influenced by root exudates and the associated soil microorganisms [16]. The plant rhizosphere plays an important role as the main reservoir of microbial diversity, and it is also considered one of the most complex ecosystems on Earth [16]. Numerous studies have demonstrated that an imbalance in rhizospheric microbial diversity leads to the occurrence of soil-borne diseases [17,18,19], such as infections with the pathogens *Rhizoctonia solani* [20], *Ralstonia solanacearum* [21], and *Fusarium oxysporum* [22], and so on, whose occurrence has been confirmed to be closely related to imbalances in rhizosphere microecology. A soil microenvironment changed by unreasonable fertilization, a deficiency in soil nutrients, and with an accumulation of root exudates resulted in the selective adaptation of the rhizosphere microbial community, enriching certain portions of the microbiome, disrupting the soil microecological balance, and finally causing plant diseases [23,24]. Therefore, soil health and a diversity of beneficial microbes are regarded as the key factors for healthy crop production and the control of diseases [25,26,27,28].

With the rapid development of bioinformatics, the changes in the microbial communities in soil have become an effective way to evaluate the occurrence of potential plant diseases [29,30,31], and metagenomics also provides us with an opportunity and a clear technical route to fully understand the interactions among plants, soil, and microorganisms [32,33]. Although several studies have been conducted to control clubroot disease in Chinese cabbage, but there are few in-depth studies about the diversity of cabbage rhizosphere microorganisms upon *P. brassicas* infection; key factors affecting the development of clubroot disease and their potential mechanisms of inhibiting clubroot disease have also not been explored.

This research sought to investigate the main factors influencing the development of clubroot disease in Chinese cabbage and the microbes that are enhanced by the pathogen’s infection of the plant. Additionally, it aimed to analyze the correlation between the pathogen, soil conditions, and the microbial community in the plant’s rhizosphere. Soil samples were obtained from various Chinese cabbage fields exhibiting four distinct levels of clubroot disease severity: high severity (HS), moderate severity (MS), mild severity (MR), and healthy (HR) soil. The properties of healthy and diseased soil were analyzed. Furthermore, metagenomic sequencing was employed to compare the rhizosphere microbial community structure and function of clubroot-infected soil. This analysis aimed to predict the onset of clubroot disease, establish a new knowledge base, and identify potential beneficial microorganisms for biocontrol purposes. 

## 2. Materials and Methods

### 2.1. Field Experiment and Sample Collection

The experiment was carried out in four agricultural plots located at Cabbage Base, Mengyang Town, Pengzhou City, Sichuan Province, China (104.0879° E, 30.9471° N). Each area was overseen using identical cultivation and agricultural practices. The Chinese cabbage cultivar selected for the study was Chuanqing 2009, which was provided by Industrial Crops Research Institute, Sichuan Academy of Agricultural Sciences. The plants were transplanted at the five-leaf stage on 15 September 2022, with a row spacing of 50 cm and a distance of 40 cm between each plant. According to our prior research and investigation, it was found that the most prevalent period of Chinese cabbage clubroot disease in Pengzhou, Sichuan Province, occurs between late November and early December. Consequently, plant samples were collected on 2 December 2022, 78 days post-transplantation. The severity of clubroot disease was assessed based on the degree of root swelling and the disease index of the cabbages plants. Based on the degree of root swelling, the collected samples were grouped as the healthy group (HR, disease index = 0), high susceptible group (HS, disease index ≥ 70), moderate susceptible group (MS, 30 ≤ disease index < 70), and mild susceptible group (MR, 0 < disease index < 30) (Figure 1A). Fifteen random cabbage plants were selected from each group, and their roots were carefully excavated. The samples were then gently gathered, excluding non-rhizosphere soil through gentle shaking, and stored in an ice bag before being transported to the laboratory within a 12-h timeframe [34,35]. The rhizosphere soil from five cabbage plants was obtained by rinsing with sterile PBS buffer and subsequently centrifuged to create a single sample. The soil samples were separated into two portions, with one portion being preserved at a temperature of −80 °C for the purpose of total DNA extraction, while the other portion was subjected to physicochemical analyses subsequent to air drying. 

### 2.2. Physiochemical Analysis

Organic matter (OM), total nitrogen (TN), available nitrogen (AN), total phosphorus (TP), available phosphorus (AP), total potassium (TK), and available potassium (AK) were extracted and analyzed according to previous reports [1,35]. The cation exchange capacity (CEC) was assessed using the ammonium saturation method. Soil pH was measured in all soil samples using a pH meter in a soil and water suspension. Based on the previously described methods, the clubroot disease index (DI) for each group was calculated as follows [24,36]:Incidence rate (%) = number of diseased plants/total number of investigated plants × 100
Disease index = (Σ (number of diseased plants at each stage × relative value))/(total number of plants under investigation × highest incidence of disease) × 100

### 2.3. DNA Extraction and Sequencing

Total DNA present in the rhizosphere soil was extracted with a Soil DNA Kit (MP Biomedicals, Solon, OH, USA). DNA concentration and quality were then measured using the Qubit dsDNA Assay Kit on a Qubit 2.0 Fluorometer (Life Technologies, Carlsbad, CA, USA) and 1% agarose gel, respectively. Metagenomic sequencing was conducted on the total DNA extracted from the rhizosphere soil using services provided by Beijing Novogene (Beijing, China). The library was constructed using approximately 1 μg of high-quality DNA. The NEBNext Ultra DNA Library Prep Kit for Illumina (New England Biolabs, Ipswich, MA, USA) was employed to generate the sequencing libraries in accordance with the manufacturer’s recommended guidelines. In summary, the DNA sample was fragmented to the size of 350 bp using sonication. Subsequently, the DNA fragments were subjected to end-polishing, A-tailing, and ligation with full-length adapters for Illumina sequencing, followed by PCR amplification. Eventually, PCR products were purified (AMPure XP system), and the libraries were assessed for size distribution through the utilization of an Agilent 2100 Bioanalyzer (Agilent Technologies, Santa Clara, CA, USA). A cBot cluster generation system was used to cluster the index-coded samples according to the manufacturer’s guidelines (Illumina Inc., San Diego, CA, USA). After cluster generation, library preparations were sequenced using the Illumina HiSeq platform (Illumina Inc.). Clean data were obtained for subsequent analysis through the initial processing of raw data obtained from the Illumina HiSeq sequencing platform. The default value was used to perform the processing steps. Reads that aligned with the Chinese cabbage genome were removed [37] using Bowtie 2.2.4 [38]. The remaining reads were utilized for subsequent metagenomic assembly and analysis.

### 2.4. Metagenomic Data Analysis

According to previously described parameters, the clean data were assembled using MEGAHIT v1.2.9 [39]. Subsequently, scaftigs containing fragments shorter than 500 bp were filtered out, and the retained scaftigs were subjected to open reading frame (ORF) prediction through the utilization of MetaGeneMark V2.10 software, with lengths below 100 nt being excluded [32]. CD-HIT version 4.8.1 [40,41] was adopted for the redundancy removal of open reading frames (ORFs) with a specified identity threshold of 95% and coverage of 90%, resulting in the generation of a unique initial gene catalog. Subsequently, Bowtie version 2.2.4 [38] was employed to acquire the unigenes by mapping clean data to the initial gene catalog. 

The abundance of individual genes in each sample was calculated according to the number of mapped reads and the length of the gene as follows: $$G_{k}=\frac{r_{k}}{L_{k}}\cdot \frac{1}{\sum_{i=1}^{n}\frac{r_{i}}{L_{i}}}$$
where *r* represents the number of reads mapped to genes and *L* represents the length of the gene. Based on the abundance of each gene in each sample, the summary statistics, core–pan gene analysis, and Venn diagram of gene counts were performed. 

### 2.5. Taxonomic Predictions and Functional Annotations

The DIAMOND v4.4.1 software [42] was adopted for aligning the unigenes to sequences of the kingdoms Bacteria, Fungi, Archaea, and Viruses. Then, the sequences were extracted from the non-redundant database of National Center for Biotechnology Information (NCBI, version: 2022.12). According to the conclusive alignment outcomes of individual sequences, system classification was performed by applying the alignment with the most minimal e-value [43] to the lowest common ancestor (LCA) algorithm through the utilization MEGAN 6 software [44], and then the species annotation data were verified. Subsequently, we acquired data pertaining to the quantity of genes and their abundance within each sample across various taxonomic levels. The total gene abundances for each species were then calculated to determine the species’ abundance within each sample. The gene count for a particular species within a given sample was equal to the quantity of genes that exhibit a non-zero level of abundance. The DIAMOND software (v0.9.9) were configured with the parameters-blastp and e-value <= 1 × 10^−5^. This setup facilitated the alignment of unigenes with the Kyoto Encyclopedia of Genes and Genomes (KEGG) Orthology (KO) profiles, the Carbohydrate-Active enZYmes (CAZy version 1.0) database, the Evolutionary Genealogy of Genes: Non-supervised Orthologous Groups (eggNOG version 5.0) database, and the Comprehensive Antibiotic Research Database (CARD version 5.10). The functional annotation data of each sample were acquired. We determined the gene abundance data of each annotated functional hierarchy, indicating the quantity of genes linked to each taxonomic hierarchy. The count of genes associated with a function having a non-zero abundance was defined as the gene count for that function in each sample. The relative abundance of each functional hierarchy was measured by summing the relative abundances of genes annotated to that specific functional level. The accuracy of the gene count table within each taxonomy hierarchy for every sample was confirmed according to the functional annotation outcomes and gene abundance data.

### 2.6. Statistical Analysis

The calculations and statistical analyses were conducted using R version 4.2.1. The ‘vegan’ package version 4.2.1 was utilized for principal component analysis (PCA), principal co-ordinate analysis (PCoA), and redundancy analysis. The Bray–Curtis distance matrix was used to perform an analysis of similarity (ANOSIM). Linear regression analysis (Pearson correlation) was carried out withing the ‘Hmisc’ package version 4.2.1. Based on the linear discriminant analysis (LDA) effect size (LEfSe), LEfSe version 1.0 software was utilized for assessing the statistical differences in taxonomic metagenomes [45]. The investigation of differential functional metagenomes was conducted using two methods: (1) Metastat analysis [46] and (2) LEfSe analysis.

## 3. Results

### 3.1. General Characteristics of Metagenome Illumina PE150 Sequencing Results

A total of 1,185,011,004 raw sequence reads from 12 shotgun metagenome libraries were obtained, with read counts per sample varying between 85,054,584 and 108,820,624. After quality-filtering and the elimination of chimeras, over 99% of the reads were deemed effective, and more than 89% of reads were preserved with a quality score of 30 or higher. This suggests that the dataset was sufficient for subsequent statistical analysis (Appendix A). Evidently,, the rarefaction curves of core and pan genes exhibited a gradual flattening, indicating that the samples encompassed a substantial portion of the microbiomes present in the rhizosphere (Appendix A). In addition, an average of 3,347,523 unigenes were identified from the 12 samples, meeting identity and coverage thresholds of over 95 and 90%, respectively. The unigene counts ranged from 3,228,418 to 3,538,232 across the four groups (Appendix A). 

### 3.2. Gene Prediction and Abundance Analysis

The Venn diagrams illustrated the unique and shared unigenes among multiple samples, providing a visual representation of the overlap of unigenes across the samples. A total of 3,600,445 unigenes were found to be common to all samples, with 3,984,705 unigenes shared between HR and HS. Additionally, HR had 68,575 unigenes that were unique to it, a significantly higher number compared to the other three samples. Conversely, the number of unigenes shared by MR, MS, and HS was 3,987,076 and 4,042,840, respectively (Figure 1B). Under the occurrence of clubroot disease, the number of genes common to all samples was increased, while the number of unique unigenes in MR, MS, and HS decreased gradually. This suggested that the occurrence of clubroot disease may impact the composition and structure of the rhizosphere soil microbial community.

### 3.3. Analysis of the Microbial Community Composition

The rhizosphere microbiota exhibited comparable distribution patterns across four groups at the kingdom level, but notable variation was observed at the phylum and genus levels (Figure 1C,D). At the kingdom level, the relative abundance of Bacteria among the four groups was approximately 90%, surpassing that of Archaea, Eukaryota, and Viruses (Figure 1C). At the phylum level, Proteobacteria, Actinobacteria, and Gemmatimonadetes exhibited a higher relative abundance across the four groups compared to other phyla, with Proteobacteria being the most prevalent. The phyla Proteobacteria and Actinobacteria were significantly more abundant in HS than in HR, while Gemmatimonadetes, Bacteroidetes, Acidobacteria, and Verrucomicrobia showed higher relative abundances in HR compared to HS (Figure 1C; Appendix A). At the genus level, the rhizosphere microbial communities primarily consisted of *Sphingomonas*, *Nocardioides*, *Gemmatirosa*, *Nitrospira*, *Lysobacter*, *Gemmatimonas*, and *Rhodanobacter* (Figure 1D). However, the dominant genera varied considerably among each group. In the HS group, *Sphingomonas* and *Rhodanobacter* exhibited the highest relative abundance. Conversely, in the HR group, the genera *Gemmatimonas*, *Gemmatirosa*, and *Lysobacter* were more prevalent compared to the other three samples. In the MR group, several genera, including *Sphingomonas*, *Nocardioides*, *Gemmatirosa*, and *Gemmatimonas*, were present at higher relative abundances than in the MS group. However, the relative abundance of *Pseudoxanthomonas* was observably lower in MR than in MS, but higher than in HS (Figure 1C; Appendix A). 

Based on the results of PCoA using Bray–Curtis distances and non-metric multi-dimensional scaling (NMDS) results, significant differentiation was observed at the genus level among the rhizosphere microbiota of the four groups (Figure 2A). Distinct differences were found between HR and HS, while MR and HR exhibited a relatively higher similarity in microbiota composition. In addition, the MS group showed an obvious separation from the HR and MR groups. Meanwhile, 56.79% of the variance was collectively explained by the PC1 and PC2 components of PCoA. The top 35 genera based on their abundance in each sample were chosen to generated a heatmap (Figure 2B), which showed that some of the rhizosphere microbial members gradually changed with the clubroot disease infection, and then clustered them from the genus level. The heatmap revealed that the genera *Sphingomonas*, *Mycobacterium*, *Rhodanobacter*, *Streptomyces*, and *Solirubrobater* were enriched in HS, whereas the rhizosphere microbiota of HR were enriched by *Gemmatimonas*, *Haliangium*, *Pedosphaera*, *Flavobacterium*, *Candidatus Nitrosotalea*, and *Gaiella*.

### 3.4. Multiple Soil Environmental Factors Significantly Correlated with Clubroot Disease

Principal component analysis revealed distinct differentiation among the samples based on soil environmental factors (Figure 3A), with HR samples being notably separated from the other three samples. Redundancy analysis was applied to clarify the correlation between the soil environmental factors and the microbial community structure. The results showed that the cation exchange capacity, organic matter, total nitrogen, and pH played a key role in structuring the microbial communities (Figure 3B). The several soil environmental factors were significantly correlated with the clubroot disease severity (Appendix A). For instance, pH, CEC, TN, and OM were significantly negatively correlated with the clubroot disease index, namely decreasing from HR to HS samples. Meanwhile, a positive but weak correlation was observed between the available phosphorus and the clubroot disease index. In addition, there were also weak correlations between AK, TK, AN, TP, and the clubroot disease index (Figure 3B). These results indicated that the clubroot disease index might have a significant impact on the microbial community structure.

Analysis conducted at the genus level indicated a significant correlation between the microbiota present in the rhizosphere soil and various soil environmental factors, as well as the severity of clubroot disease (Figure 3C). The results suggested that 23 genera exhibited a negative correlation with the clubroot disease index, whereas 27 genera showed a positive correlation. Among them, *Polaromonas* demonstrated an exceptionally significant positive correlation with the clubroot disease index (*p* < 0.01), with *Rhodanobacter*, *Burkholderia*, *Paraburkholderia*, and *Bordetella* also displaying significant positive correlations with the clubroot disease index (*p* < 0.05). In contrast, *Candidatus Solibacter*, *Ramlibacter*, and *Bryobacter* exhibited a notably strong negative correlation with the clubroot disease index (*p* < 0.01), while *Thauera*, *Caenimonas*, and *Curvibacter* displayed a significant negative correlation with the clubroot disease index (*p* < 0.05). Among the top 10 genera, *Gemmatirosa* and *Gemmatimonas* had a higher correlation with the available nitrogen compared with other species, whereas *Rhodanobacter* was more affected by the organic matter, pH, and total nitrogen compared with other species. In addition, the organic matter contents showed a substantial impact on *Pseudomonas*.

### 3.5. Changes in Core and Specific Microbes after Pathogen Infection

Principal component analysis revealed that core microbial species were able to effectively differentiate between various samples (Figure 4A), with HR samples being distinct from HS samples. LEfSe analysis (LDA score > 4.0, *p* < 0.05) determined the important core microbes of each group. From the phylum level to the species level, different clubroot disease grades had specific effects on the microbial communities. As shown in Figure 4B, HR and HS had 23 and 16 enriched taxa, respectively. The HR group significantly enriched six microbial taxa encompassing the phyla Gemmatimonadetes and Candidatus Rokubacteria (from phylum to genus), Candidatus Eisenbacteria (from phylum to family), the order Glycomycetales and Thaumarchaeota, and the species *Candidatus Nitrosotalea devanaterra* (Figure 4B,C). However, a significant proportion of taxa within the genus *Sphingomonas* and *Rhodanobacter* were recruited by HS (Figure 4B,C). Similar fundamental organisms were also identified in both HS and MR samples, as shown in Appendix A. Conversely, the phyla Calditrichaeota and Candidatus Wallbacteria were prominent in HR, whereas the phyla Candidatus Lokiarchaeota and Candidatus Delongbacteria were enriched in HS (Appendix A). In addition, the order Pseudonocardiales was also prominent in HS. Furthermore, the species belonging to Proteobacteria (e.g., *Polaromonas*) were more abundant in the MS and HS groups than in the MR and HR groups. In addition, substantial differences were observed from the core taxa in HR compared with other samples, and multiple taxa within the order Glycomycetales and the genus *Gemmatimonas* were enriched in HR (Appendix A).

### 3.6. Functional Composition of Rhizosphere Microbiota

A total of 2,270,910 unigenes from all samples were obtained, attributing to six categories of gene functions and 386 KEGG pathways. The majority of unigenes were related to metabolism, comprising 62.7% of the total (Figure 5A). Additionally, glycoside hydrolases were the most prevalent among carbohydrate enzymes, representing the largest proportion of unigenes at 43.4% (Figure 5B). Furthermore, a collective sum of 6836 KOs was detected across the samples, with metabolism-related pathways being the most prevalent. These pathways constituted approximately 18.6% to 19.1% of the identified KOs, representing the largest proportion within the dataset (Appendix A). At KEGG Level 2, the primary functional categories identified were pathways correlated with the metabolism of carbohydrates, amino acids, energy, cofactors, and vitamins (Appendix A). Upon further examination at KEGG Level 3, it was found that purine metabolism (ko00230) was the most prominent pathway, followed by ABC transporters (ko02010), two-component systems (ko02020), oxidative phosphorylation (ko00190), quorum sensing (ko02024), pyrimidine metabolism (ko00240), pyruvate metabolism (ko00620), carbon fixation pathways in prokaryotes (ko00720), ribosomes (ko03010), and glyoxylate and dicarboxylate metabolism (ko00630; Appendix A). PCoA utilizing Bray–Curtis distances revealed a distinct separation of the HR group from the remaining three groups, which clustered together (Appendix A–C). Further analysis was taken to annotate gene functions using the eggNOG database. The results suggested that “function unknown” was the predominant function among all categories at Level 1, followed by amino acid transport/metabolism and energy production/conversion functions (Appendix A). The ABC transporter, histidine kinase, and dehydrogenase functions were more prevalent than other categories at Level 2 (Appendix A). Furthermore, ENOG410XNMH histidine kinase, serine threonine protein kinase (COG 0515), acriflavin resistance protein (COG0841), acyl-CoA dehydrogenase (COG1960), dehydrogenase (COG1012), two-component, sigma54 specific, transcriptional regulator, Fis family (COG2204), AMP-dependent synthetase/ligase (COG0318), aldehyde oxidase/xanthine dehydrogenase, molybdopterin binding (COG1529), the ABC transporter (COG1131), and the phosphorelay signal transduction system (COG1028) were identified (Appendix A). PCoA utilizing Bray–Curtis distances of function annotation abundance indicated that MR, MS, and HS exhibited clustering at Levels 1 and 2, as well as in orthologous groups (OGs) (Appendix A). Conversely, HR was clearly separated from the aforementioned three groups. These patterns were consistent with the findings observed in the KEGG analysis (Appendix A).

### 3.7. Representative Microbial Functions after Pathogen Infection

The predominant core functions within the four groups were identified through the utilization of Metastats analysis and LEfSe analysis (LDA score > 2.5, *p* < 0.05). Within the KEGG ortholog groups, there was a notable enrichment of those linked to glycerol-1-phosphatase (K06116) in the HR and MR groups, with a particularly significant disparity observed between the MR and HS groups (Figure 6A). At KEGG Level 1, pathways related to environmental information processing were significantly enhanced following the invasion of *P. brassicae*, but showed a decline as the severity of the disease increased (Figure 6B). This trend was also observed in seven pathways at a more detailed level, including those involved in signal transduction, infectious diseases, cell motility, the immune system, and development. (Appendix A). At KEGG Level 3, the pathways of ether lipid metabolism (ko00565) and the circadian rhythm (ko04710) were revealed to be more prevalent in the HS context compared to the HR context (Figure 6C,D). However, oxidative phosphorylation (ko00190) was uniquely enriched in the HR condition (Figure 6D).

### 3.8. Analysis of Rhizosphere Microbial Antibiotic Resistance Ontologies (AROs)

A total of 564 AROs were identified within the rhizosphere soil samples of Chinese cabbage, exhibiting an abundance spanning from 0.6779 ppm to 925.8986 ppm. These AROs were linked to a total of 30 distinct antibiotic categories, with macrolides (15.60%), aminoglycosides (15.60%), cephalosporins (13.65%), carbapenems (7.62%), and glycopeptides (7.09%) being the most prevalent types among them (Appendix A). The macrolides were mainly msrB, oleB, and LpeB, whereas the aminoglycoside were mainly *APH6-Ia*, *AAC6-It*, and *aadA5*. Among the detected AROs, *adeF* had the highest abundance (10.091%), followed by *tet45* (2.588%) (Figure 7A). LEfSe analysis (LDA score > 2.5, *p* < 0.05) determined the antibiotic resistance genes with significant differences in each group. As shown in Figure 7C, there were eight, five, seven, and seven AROs enriched in HR, HS, MR, and MS, respectively. The heatmap analysis demonstrated an enrichment of the antibiotic resistance genes *tet45*, *PmpM*, and *rpoB2* in HR, while the antibiotic resistance genes *MexK*, *emrB*, *tmrB*, and *MexI* were enriched in HS (Figure 7D). Various mechanisms of antibiotic resistance were examined, with antibiotic inactivation being the most prevalent (38.26%), followed by antibiotic efflux (36.36%), antibiotic target alteration (18.56%), and antibiotic target protection (3.03%) (Figure 7B).

## 4. Discussion

Clubroot disease is a prevalent soil-borne ailment that significantly impacts the yield and quality of Chinese cabbage, presenting a notable challenge in agricultural production [47]. Rhizosphere microbial composition and structure exert a vital role in maintaining soil health and increasing plant disease resistance [18]. When pathogens infiltrate plant roots, it causes a disruption in the balance of the soil ecosystem, which subsequently leads to changes in the population and variety of microbial communities in the rhizosphere. Hence, it is imperative to adopt a methodical approach to comprehend the interplay between the soil environment, pathogens, plants, and rhizosphere microbes, as this is crucial for fostering the optimal growth of crops.

This research aimed to examine the variations in soil environments and microbial attributes in the rhizosphere soil of Chinese cabbage plants infected with clubroot disease of varying severities. This was achieved through the utilization of shotgun metagenomic sequencing. Additionally, the study evaluated the potential correlations among soil conditions, microbial communities, and the severity of clubroot disease.

### 4.1. Response of Microbial Composition and Structure to Clubroot Disease

In the rhizosphere soil of Chinese cabbage, the microbial community exhibited resemblances among different levels of clubroot disease at the kingdom level. Nonetheless, there was a notable variation in the composition of rhizosphere microbes across different classes, indicating a significant association between the distribution of rhizosphere microbiota in Chinese cabbage and the severity of clubroot disease. The variation in rhizosphere microbial diversity among distinct groups was notably evident at both the phylum and genus levels. Analysis revealed that the HS soil exhibited a greater prevalence of the phyla Proteobacteria and Actinobacteria compared to the HR soil. Conversely, an elevated relative abundance of Gemmatimonadetes, Bacteroidetes, Acidobacteria, and Verrucomicrobia was observed in the rhizosphere of cabbage soils from the HR group. This finding aligns with a recent study that observed a notable disparity in the diversity of rhizosphere microbes between Chinese cabbage plants infected with clubroot disease and those that were healthy [48]. Moreover, there were notable variations in the predominant genera observed among groups exhibiting varying levels of clubroot disease severity. The HR group exhibited a higher presence of distinct genera compared to the other groups, such as *Gemmatimonas*, *Gemmatirosa*, and *Lysobacters*, which showed a decreased relative abundance in the HS group. Moreover, there were 23 genera that exhibited a negative correlation with the clubroot disease index, while 27 genera showed a positive correlation. Several genera that exhibited a negative correlation, such as *Gemmatimonas* [49], *Pseudoxanthomonas* [50], and *Pseudomonas* [51], have been proven to be beneficial for inhibiting soil-borne diseases. Furthermore, the relative abundance of *Sphingomonas* was notably higher in the HS group, likely due to its recruitment as a beneficial microorganism to shield Chinese cabbage from disease pathogens. These findings align with prior research indicating that the levels of beneficial microbiota, especially *Sphingomonas* and *Pseudomonas*, were elevated in *Ralstonia solanacearum*-infected cucumbers [21]. In addition, a majority of genera exhibited a greater relative abundance in MR compared to MS, including *Sphingomonas*, *Nocardioides*, *Gemmatirosa*, and *Gemmatimonas*. These genera hold promise for potential biocontrol agent development against clubroot disease, pending assessment of their colonization efficacy and inhibitory impacts in field conditions. 

Principal coordinate analysis on the basis of Bray–Curtis distances and non-metric multi-dimensional scaling were adopted for evaluating the differences in microbial community structure across four groups. The clustering analysis indicated a notable disparity in the microbial community structure between the HR and HS groups, suggesting different microbial structures in the HR and HS soil samples. Moreover, the MR soil did not exhibit distinct differentiation from the HR soil, but it did show a clear distinction from the MS soil. This indicates that the composition of the microbial community in the soil was altered by the presence of clubroot disease. Previous research has also demonstrated significant differences in the bacterial community between healthy soil and soil infected with clubroot disease [52], which aligns with the findings of the current study. The varying distributions of these microbial classes in rhizosphere soils affected by different levels of clubroot disease indicate their potential resistance to the disease. This suggests that they could be utilized in the management and control of clubroot disease. 

### 4.2. Differential Microbes and Their Relationship with Soil Physiochemical Properties

The high presence of microorganisms in the rhizosphere soil is associated with their ability to resist clubroot disease, a resistance that can be enhanced through greater microbial diversity. Research has shown that the activity of the microbiome in the rhizosphere is enhanced with a greater variety of probiotics, leading to an improved suppression of plant diseases through increased competition for resources and interference with pathogens [53]. A previous field trial revealed that the composition of rhizosphere soil microbes was significantly notably altered, resulting in enhanced microbial diversity following the implementation of a biological control agent, and therefore, the damage induced by clubroot was mitigated [54]. Environmental factors, including temperature, humidity, illumination, and soil physiochemical characteristics, greatly affect the interactions between plants, pathogens, and biocontrol agents. These interactions can lead to physiological changes in host plants and affect the development of diseases [55]. Hence, the recognition of distinct microorganisms has the potential to isolate resistant strains and offer potential solutions for controlling clubroot disease. 

In a previous study, three dominant strains were isolated from the clubroot disease-infected rhizosphere soil, and subsequently employed as biocontrol agents for Chinese cabbage clubroot disease. Dramatic biocontrol and promotion effects were acquired from those two strains, which significantly improved the growth and resistance of Chinese cabbage by suppressing the incidence and severity of clubroot disease [56]. In this study, the results revealed a positive correlation between the prevalence of Sphingomonas in the rhizosphere soil and soil pH, potentially attributed to its predominant presence in alkaline soil conditions. *Sphingomonas* species play a crucial role in abiotic stress tolerance, pathogen suppression, and plant growth promotion [57,58]. In clubroot disease-infected soil, the genus *Pseudomonas* exhibited the highest relative abundance in MR, but decreased in MS and HS. This finding aligns with prior research indicating that Pseudomonas may play a role in pathogen suppression and have significant inhibitory effects on clubroot disease [59]. These results suggest that *Pseudomonas* has the potential to serve as an indicator of the presence and severity of clubroot disease. Another study demonstrated that the presence of *Pseudomonas* could improve plant health and yield production due to its beneficial metabolites, which played an important role in the biocontrol of plant disease [60]. Similar to *Pseudomonas*, the *Nocardioides* genus also plays a crucial role in the biocontrol of various plant diseases, positioning it as a promising candidate biocontrol reagent and biofertilization strategy [61]. Previously, the biocontrol efficacy of *Heteroconium chaetospira* on clubroot disease was significantly impacted by soil pH, which was recognized as a key factor along with moisture levels and pathogen concentration [62]. Herein, a positive correlation was observed between the prevalence of *Nocardioides* in the rhizosphere soil and the soil pH. This suggests that plants have the ability to attract advantageous microorganisms in response to pathogen presence, thereby potentially bolstering resistance against soil-borne diseases. Future efforts in developing bioengineering microbes based on *Nocardioides* should prioritize considerations of pH levels. 

The plant performance and their corresponding microbial communities were directly affected by alterations in soil characteristics in response to shifting environmental conditions [63]. A previous study demonstrated that soil AP played a crucial role in influencing the composition of plant-associated microbiomes [52]. The RDA analysis conducted in this study established significant negative correlations between the clubroot disease index and pH, CEC, TN, and OM, which were also significantly associated with rhizosphere soil microbes. This result was consistent with previous studies in which a higher pH level could reduce root hair infection of *P. brassicae* [64]. Moreover, there was a notable positive relationship between *Gemmatimonas* and nitrogen availability, while a negative association was observed with the severity of clubroot disease. A prior investigation indicated that certain strains of *Gemmatimonas* quickly established themselves in a restricted ecological niche within the rhizosphere of soil affected by *Fusarium* wilt disease, consequently mitigating the damage caused by *Fusarium* wilt [65]. In addition, the presence of organic matter significantly influenced the activity of *Pseudomonas*. Previous research has demonstrated that plants have the ability to recruit stress-resistant microbes from the environment and shape their rhizosphere microbial communities [66]. Therefore, in response to pathogen invasion, plants recruit advantageous microorganisms from the rhizosphere through self-regulation to enhance their resistance to diseases. Nevertheless, the intricate processes that regulate the assembly of plant microbiomes in response to varying growth environments and their management are highly complex and somewhat unpredictable. It is essential to isolate potentially beneficial microbes and assess their efficacy in suppressing diseases through in vivo experimentation.

### 4.3. Response of Microbial Functional Adaptation to Clubroot Disease

The functions of the rhizosphere microbiota were also analyzed in addition to their composition. In recent reports, genes involved in plant-microbiome signaling pathways were enriched in the *Fusarium* wilt-diseased root endosphere of chili pepper (*Capsicum annuum* L.) [66]. These microbial functions were revealed to play crucial roles in enhancing plant resistance against pathogen infections [67]. In the present study, metabolism accounted for the greatest number of functions, being the predominant function no matter whether in healthy Chinese cabbage plants or those with different grades of clubroot disease. Within KEGG ortholog groups, those associated with glycerol-1-phosphatase were enriched in the HR and MR groups, with an extremely significant difference being observed between the MR and HS groups. In a previous study, *Aureobasidium pullulans* was found to enhance glycerolipid metabolism, resulting in the buildup of elevated levels of glycerol as a mechanism to withstand heightened osmotic stress. The organism achieved a harmonious equilibrium between cell growth and acclimatization to salt stress through the regulation of glycerol accumulation and utilization [68]. The enhancement of disease resistance in plants following treatment with glycerol was attributed to the induction of systemic acquired resistance, leading to the suppression of brown rot in peach fruit. It was hypothesized that this effect may be linked to the enhancement of polyunsaturated fatty acid synthesis [69]. Moreover, oxidative phosphorylation was uniquely enriched in HR, serving as a pivotal pathway in metabolic processes. A recent study indicated that ketogenesis can enhance oxidative phosphorylation, thereby enhancing the body’s ability to tolerate diseases [70]. Further, environmental information processing-associated pathways were extremely strengthened after the invasion of *P. brassicae*, but weakened as the severity of the disease increased. This observation could potentially support the concept of a plant’s response akin to a "cry for help" when confronted with pathogenic invasion, although culture-based experiments are necessary to confirm this hypothesis. Collectively, the regulation of pathways linked to glycerolipid metabolism and oxidative phosphorylation could serve as crucial mechanisms employed by rhizosphere microorganisms to alleviate the detrimental impacts of pathogen infection on Chinese cabbage.

The abuse of antibiotics leads to irreversible changes in microbial communities within the environment, presenting potential hazards to both human health and the ecological balance. Consequently, there has been a significant focus among researchers on studying resistance genes [71]. For instance, studies have identified the presence of daptomycin and colistin in the rumen of sheep [72], as well as elevated levels of tetracycline and erythromycin in the fecal matter of pigs, chickens, and humans [73].

In the annotation of antibiotic resistance genes, it was observed that the predominant phyla harboring these genes in the species were consistent with those identified in the annotation of the samples, specifically Proteobacteria, Actinobacteria, Acidobateria, and Gemmatimonadetes. Proteobacteria exhibited a higher prevalence of antibiotic resistance genes compared to other bacterial groups. The antibiotic resistance genes *tet45*, *PmpM*, and *rpoB2* were found to be more abundant in HR, while *MexK*, *emrB*, *tmrB*, and *MexI* were enriched in HS. Previous research has indicated that mechanisms of antibiotic resistance linked to antibiotic inactivation are prevalent, with varying antibiotic resistance genes observed in different microbial environments [74,75]. Therefore, it is hypothesized that the severity of clubroot disease significantly impacts the abundance of antibiotic resistance genes present in individual samples.

## 5. Conclusions

In this study, the severity of clubroot disease had a notable impact on both the composition and structure of the microbial community in rhizosphere soil. Moreover, the functionality of the microbial community was markedly affected by the disease severity. The findings revealed four distinct microbial species, namely *Pseudomonas*, *Gemmatimonas*, *Sphingomonas*, and *Nocardioides*, which exhibited promising potential for the biocontrol of clubroot disease. Soil pH, organic matter contents, total nitrogen, and cation exchange capacity were the major environmental factors modulating plant microbiome assembly. In addition, microbial environmental information processing was extremely strengthened when the plant was subjected to pathogen invasion, but weakened when the severity of the disease increased. In particular, oxidative phosphorylation and glycerol-1-phosphatase might play important roles in enhancing Chinese cabbage’s resistance to clubroot disease. This work revealed the interactions and potential mechanisms among Chinese cabbage, soil conditions, clubroot disease, and the structure and functions of microbial communities. These findings could serve as a valuable basis for future research utilizing microbiological or metabolic approaches to develop disease-resistant cultivation techniques.

## Figures and Tables

**Figure 1 microorganisms-12-01370-f001:**
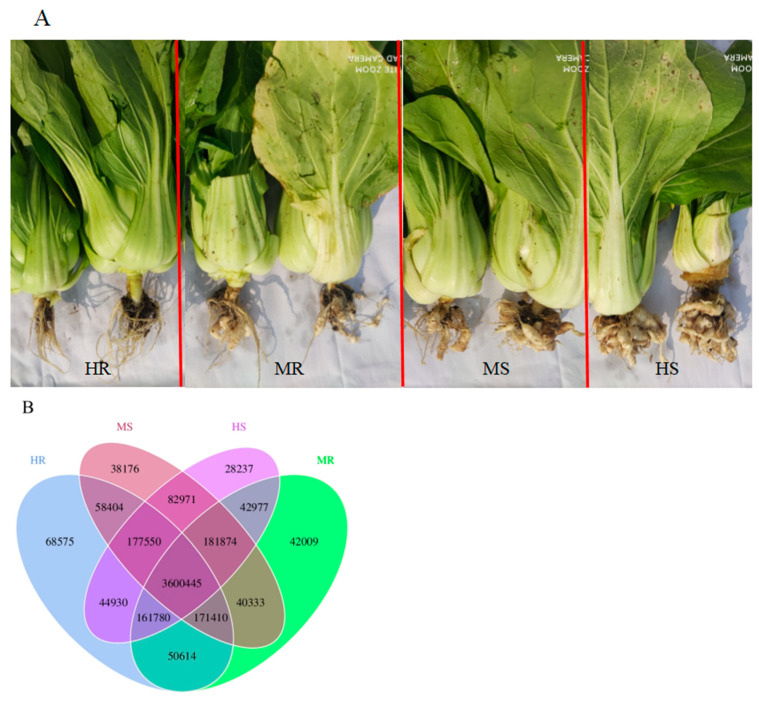
Rhizosphere soil microbiota of four groups. (**A**) The occurrence of clubroot disease in Chinese cabbage from four groups; (**B**) Venn diagram of the difference in the unigenes number between samples; (**C**) Microbial community composition at the kingdom, phylum, and genus levels; (**D**) Relative abundance of the Chinese cabbage rhizosphere soil microbial communities at the phylum and genus levels under four different clubroot disease grades.

**Figure 2 microorganisms-12-01370-f002:**
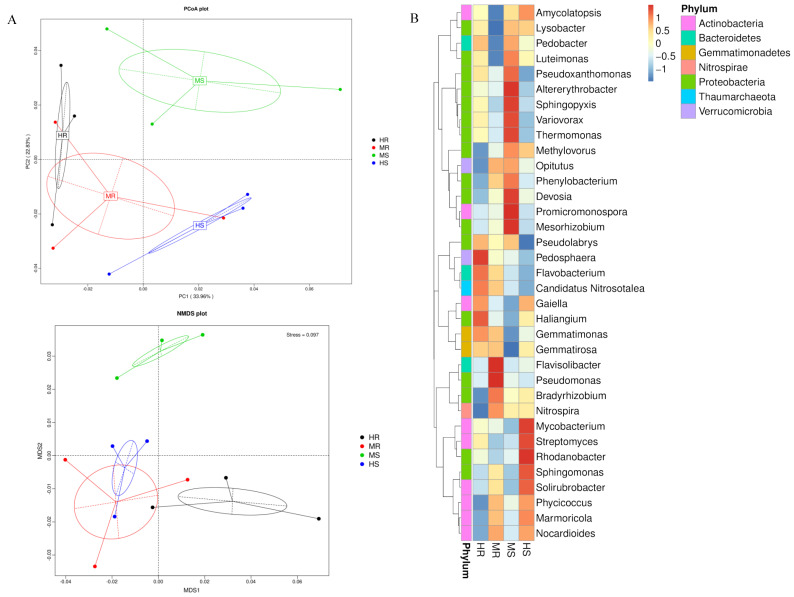
Dimension reduction analysis and clustering heatmap based on the abundance of microbiota. (**A**) Principle co-ordinate analysis based on Bray–Curtis distances (PCoA) and non-metric multi-dimensional scaling (NMDS) of microbiota descriptions in different groups at the genus level; (**B**) Heatmap of the relative abundance of 35 microbial genera.

**Figure 3 microorganisms-12-01370-f003:**
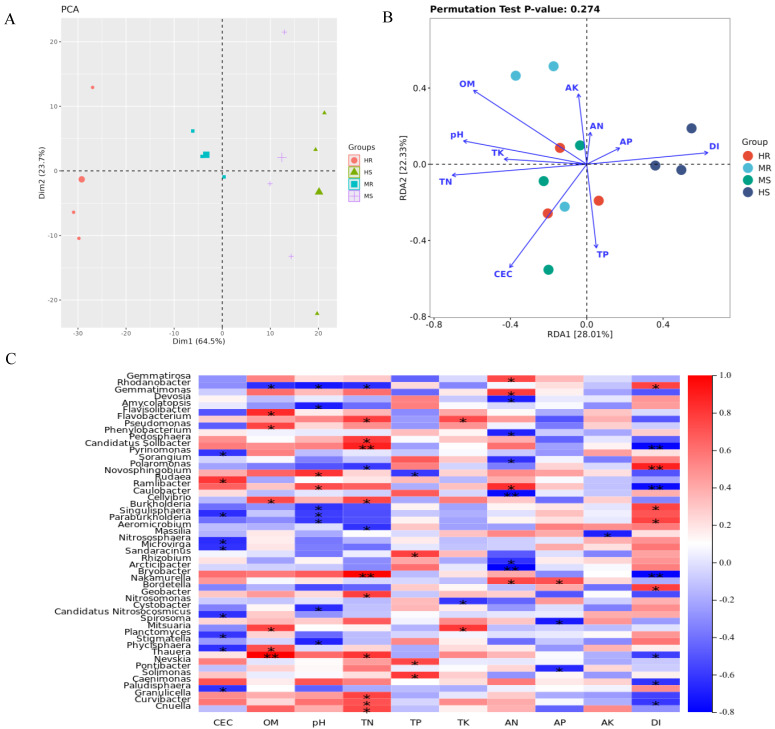
Correlation and differences among soil environmental factors and redundancy analysis of the Chinese cabbage rhizosphere soil microbial communities at the genus level under different clubroot disease grades. CEC, cation exchange capacity; OM, organic matter; TN, total nitrogen; TP, total phosphorus; TK, total potassium; AN, available nitrogen; AP, available phosphorus; AK, available potassium; DI, disease index. * denotes significance *p* < 0.05. ** represents significance *p* < 0.01. (**A**) Principal component analysis (PCA) based on soil environmental factors; (**B**) Redundancy analysis (RDA) of the microbial community; (**C**) Pearson correlation coefficient diagram between genera and soil environmental factors.

**Figure 4 microorganisms-12-01370-f004:**
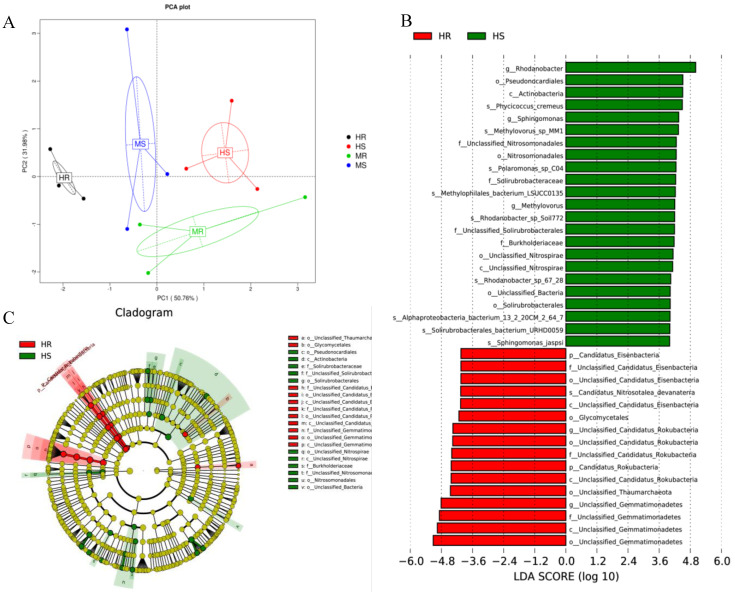
Linear discriminant analysis effect size of the Chinese cabbage rhizosphere microbial biomarkers under different under different clubroot disease grades. (**A**) Principal component analysis (PCA) based on microbial species with significant differences; (**B**) LDA scores of biomarker microbiomes; (**C**) Microbial core community composition.

**Figure 5 microorganisms-12-01370-f005:**
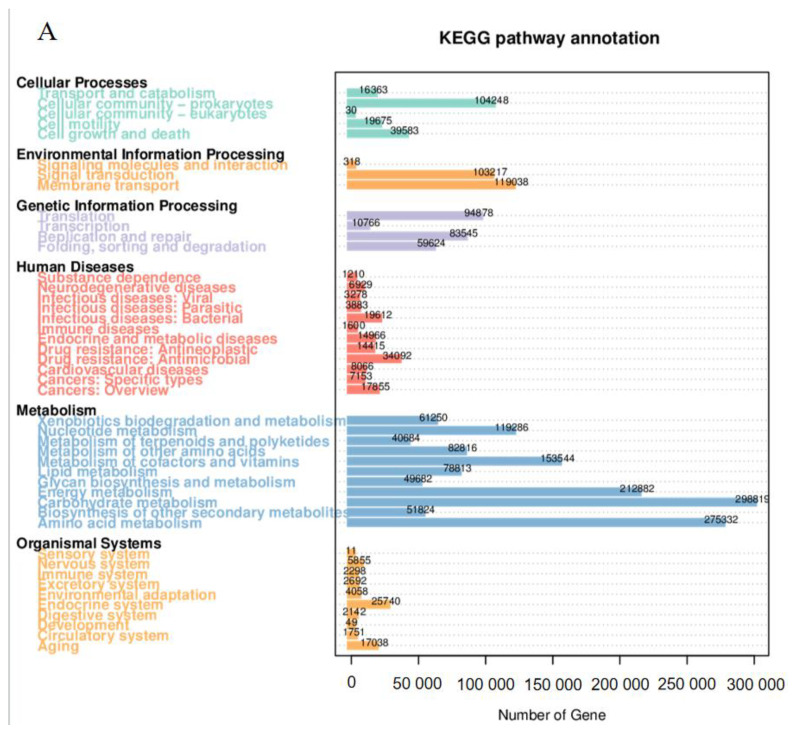
Statistical charts for the number of annotation genes. (**A**) The number of annotation genes in six metabolic functions using KEGG pathways; (**B**) The number of annotation genes in six functional carbohydrate enzymes using the CAZy database.

**Figure 6 microorganisms-12-01370-f006:**
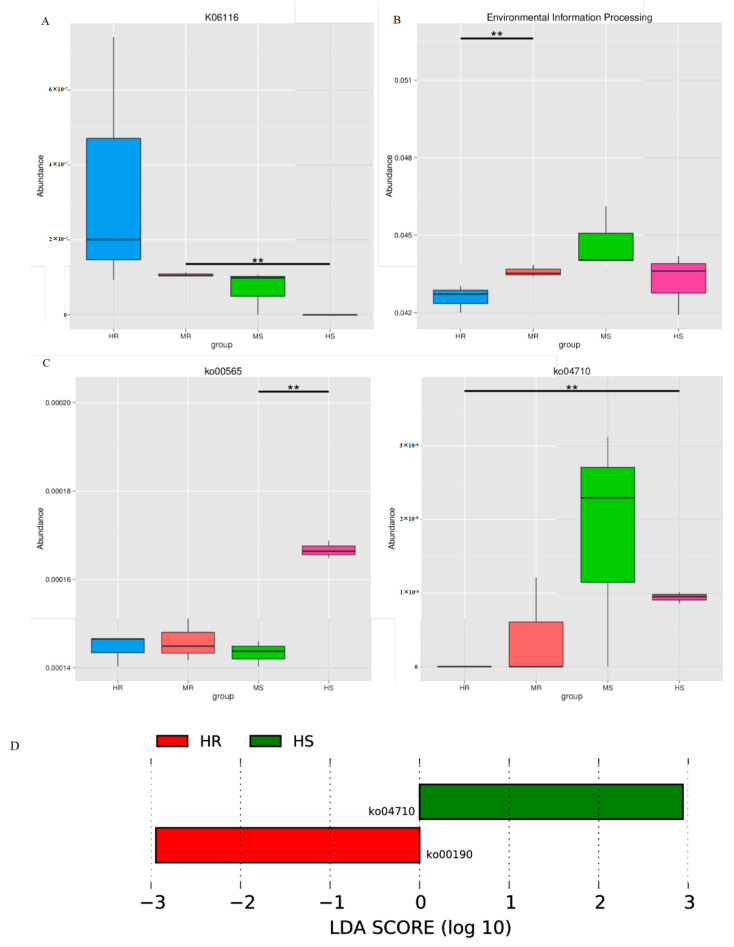
Box diagrams of functions with a significant difference using Metastats analysis. ** represents significance *p* < 0.01. (**A**) The relative abundance of differential KOs; (**B**,**C**) The number of distinct functions at Levels 1 and 3 using KEGG pathways; (**D**) The functions with a notable discrepancy at Level 3 using LEfSe analysis.

**Figure 7 microorganisms-12-01370-f007:**
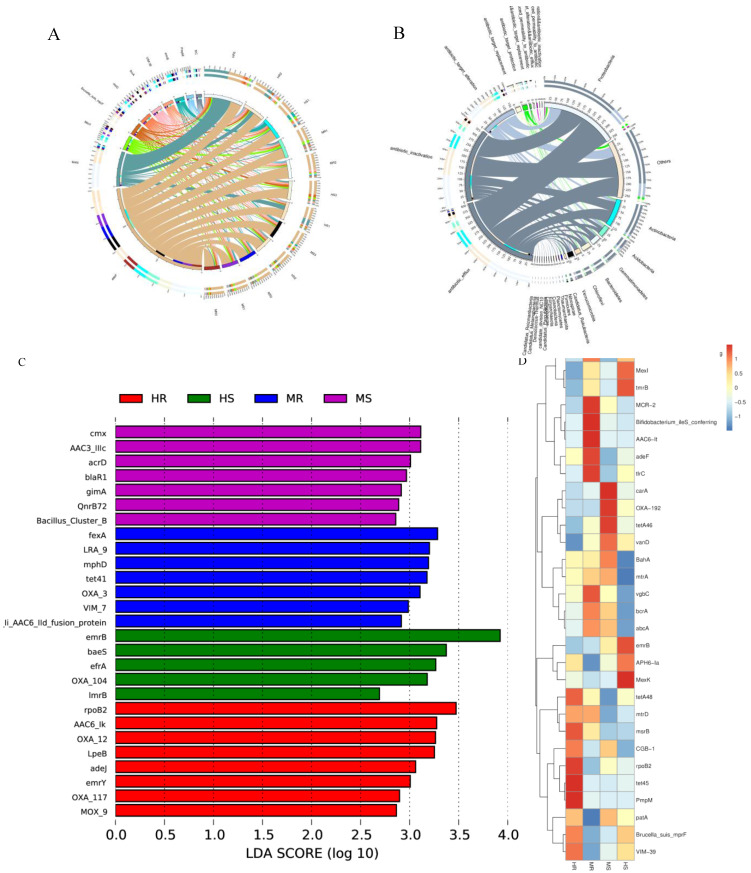
AROs in the rhizosphere of Chinese cabbage. (**A**) Overview circle diagram of relative abundance of the top 10 AROs; (**B**) Overview circle diagram of antibiotic resistance mechanisms and microbial species; (**C**) The AROs with substantial differences using LEfSe analysis; (**D**) Heatmap of the relative abundance of the top 30 AROs.

## Data Availability

The original contributions presented in the study are included in the article/Appendix A, further inquiries can be directed to the corresponding authors.

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
