# Peer review of "Comparative Metagenomic Analysis Reveals Rhizosphere Microbiome Assembly and Functional Adaptation Changes Caused by Clubroot Disease in Chinese Cabbage"

_microorganisms, 2024, doi:10.3390/microorganisms12071370_

Round 1

Reviewer 1 Report

Comments and Suggestions for Authors

Liu et al present a very interesting manuscript in which they studied the microbiota from rizospheric soils of Chinese cabbage crops affecting by  the pathogen Plasmodiophora brassicae. The authors also explore key factors affected by clubroot disease , also the stablished the relationship between environmental factors and rhizosphere microbial community. 

In my opinion, the study's design is quite successful, and has as an additional merit that was sample  soil with  four different types of pathogenic burden.  The outcomes from this study could unveil  a novel foundation and develop for efficient biocontrol agents .

The discussion addresses very satisfactory fashion  the results obtained by the authors and allows to clearly establish the relationship with the conclusions presented

in my opinion the manuscript present the quality to be published in micro-organism 

Comments on the Quality of English Language

English is clear and easy to follow 

Author Response

Thank you for your valuable comments. I revised them one by one according to your suggestion.

Reviewer 2 Report

Comments and Suggestions for Authors

This paper reports the rhizosphere microbiota of the clubroot disease infected with different severity by employing metagenomic sequencing.  The authors reveal that clubroot disease severity significantly affected microbial community composition, structure of rhizosphere soil, and microbial functions. The findings of this paper are interesting and should provide useful information to readers of Microorganisms. However, the following points need to be improved before this paper can be accepted

1. P1, Introduction, LL2-3, “which has been --- among vegetables [1]”: Cabbage is thought to have been first grown by the Greeks over 3,000 years ago, but the history of Chinese cabbage cultivation is much shorter. The authors should confirm this point. For example, Kalloo and Rama (see below) described that Chinese cabbage has been in cultivation since the fifth century.

G. KALLOO, M.K. RANA,

10 - Chinese cabbage: Brassica pekinensis, B. chinensis,

Editor(s): G. KALLOO, B.O. BERGH,

Genetic Improvement of Vegetable Crops,

Pergamon,

1993,

Pages 179-186,

ISBN 9780080408262,

https://doi.org/10.1016/B978-0-08-040826-2.50014-X.

2. P3, Materials and Methods, Field experiment and sample collection, L3, “The Chinese cabbage”: The authors should describe information on cabbage varieties and seed suppliers.

3. P3, Materials and Methods, Field experiment and sample collection, LL8-10, “Based on root swelling, samples collected were grouped as: healthy group (HR, disease index = 0), high susceptible group (HS, disease index ≥ 70%), moderate susceptible group (MS, 30% ≤ disease index < 70%) and mild susceptible group (MR, 0 < disease index < 30%)”: What does “%” mean? Size? Weight? Please describe how the authors determined “%”.

4. P15, Differential microbes and their relationship with soil physiochemical properties, 2nd paragraph, “three dominant strains were isolated from the clubroot disease infected rhizosphere soil, which were used as the biocontrol agents for the Chinese cabbage clubroot disease. Dramatic biocontrol and promotion effects were acquired from those two strains, which significantly improved the growth and resistance of Chinese cabbage by suppressing the incidence and severity of the clubroot disease [58]”. Which is correct, “three strains” or “two strains”?

5. P16, Response of microbial functional adaptation to clubroot disease, L17, “oxidative phosphorylation was stimulatet by ketogenesis,”: “stimulatet” should be “stimulated”.

6. The authors sampled plants 78 days after transplanting, but was there any clubroot decay? If there was, it should have affected the rhizosphere microbiome. This point should be briefly described in the text.

Author Response

(The authors gave the same response as above.)
